# Hierarchical Ultrathin Layered GO-ZnO@CeO_2_ Nanohybrids for Highly Efficient Methylene Blue Dye Degradation

**DOI:** 10.3390/molecules27248788

**Published:** 2022-12-11

**Authors:** Marimuthu Karpuraranjith, Yuanfu Chen, Ramadoss Manigandan, Katam Srinivas, Sivamoorthy Rajaboopathi

**Affiliations:** 1School of Integrated Circuit Science and Engineering, State Key Laboratory of Electronic Thin Films and Integrated Devices, University of Electronic Science and Technology of China, Chengdu 610054, China; 2School of Science and Institute of Oxygen Supply, Tibet University, Lhasa 850000, China; 3Department of Chemistry, Government Arts College for Women, Sivagangai 630561, India

**Keywords:** GO-ZnO@CeO_2_, nanostructures, interfacial contact, suitable band-gap matching, photocatalytic degradation

## Abstract

Highly efficient interfacial contact between components in nanohybrids is a key to achieving great photocatalytic activity in photocatalysts and degradation of organic model pollutants under visible light irradiation. Herein, we report the synthesis of nano-assembly of graphene oxide, zinc oxide and cerium oxide (GO-ZnO@CeO_2_) nanohybrids constructed by the hydrothermal method and subsequently annealed at 300 °C for 4 h. The unique graphene oxide sheets, which are anchored with semiconducting materials (ZnO and CeO_2_ nanoparticles), act with a significant role in realizing sufficient interfacial contact in the new GO-ZnO@CeO_2_ nanohybrids. Consequently, the nano-assembled structure of GO-ZnO@CeO_2_ exhibits a greater level (96.66%) of MB dye degradation activity than GO-ZnO nanostructures and CeO_2_ nanoparticles on their own. This is due to the thin layers of GO-ZnO@CeO_2_ nanohybrids with interfacial contact, suitable band-gap matching and high surface area, preferred for the improvement of photocatalytic performance. Furthermore, this work offers a facile building and cost-effective construction strategy to synthesize the GO-ZnO@CeO_2_ nanocatalyst for photocatalytic degradation of organic pollutants with long-term stability and higher efficiency.

## 1. Introduction

In recent decades, global conservational circumstances have changed with rapid world development, creating new research topics. There has been an increase in the growth of several industries, such as automobile, electronic, furniture and textile dyeing production. Previously, manufacturing companies discharged color and colorless pollutants into wastewater. Water treatment has widespread research opportunities to limit this. Finding a simple, cost-effective and highly efficient route to eliminate industrial contaminants from wastewater is an exciting environmental research topic [1,2].

The chosen semiconducting materials have more significant factors that can influence the photo-catalytic reaction of nanomaterials, such as high electron–hole pair lifetime, being non-toxic, nanostructure, more active surface area, interface contact and suitable band gap. Generally, metal oxide and metal chalcogenide nanomaterials, such as cerium oxide (CeO_2_), iron oxide (Fe_2_O_3_), titanium oxide (TiO_2_), tin oxide (SnO_2_), zinc oxide (ZnO), tin sulfide (SnS) and zinc sulfide (ZnS) are the most widespread semiconducting nanomaterials that have been used in catalytic reactions previously [3,4,5]. These semiconducting catalysts have some restrictions on the improvement of catalytic activity under visible light irradiation. The ZnO and TiO_2_ nanomaterials have a wide-band energy gap in the UV region. It can be assumed that approximately 10% of solar energy will be absorbed with these materials, making them not suitable for practical applications [6].

Nowadays, rare-earth metal oxides have been established. Cerium (Ce) typically has two different forms of oxide: cerium sesquioxide (Ce_2_O_3_) and cerium dioxide (CeO_2_), respectively. Moreover, CeO_2_ has fluorite structure (FCC) and higher stability than Ce_2_O_3_. Cerium belongs to a rare-earth family and is more abundant than tin and copper [7,8]. High abundance makes this material scientifically significant, with an extensive range in several applications such as biotechnology, oxygen permeation membrane systems, environmental chemistry, fuel cells, glass-polishing materials, low-temperature water–gas shift reaction, auto-exhaust catalysts, oxygen sensors and electrochromic thin-film applications, as well as photocatalysts and medicine [9,10].

Typically, the photocatalytic activity of rare-earth family materials is enhanced by doping constituents for modification of the energy band gap. Nevertheless, such doping constituents characteristically produce the subordinate pollutants [11,12]. Hence, they are frequently attached to another semiconducting material that can increase the electron–hole pair lifetime. Recently, several studies have examined the effectiveness of combining carbon with metal oxides, for example, carbon nanotubes (CNTs), graphene oxide (GO) and reduced graphene oxide (rGO), as an effective way to decrease the electron–hole pair recombination rate and avoid the accumulation of metal oxide particles [13,14,15].

Therefore, various combined metal oxides, such as TiO_2_@GO, TiO_2_@CNT ZnO@RuO_2_@rGO and ZnO@CNT, offer great photocatalytic performance while reducing the recombination rate and spreading the energy range in which photoexcitation occurs. However, the mechanism and catalytic effectiveness underlying graphene oxide-ZnO nanosheet activity are not well understood [16,17,18,19]. We think that it is reasonable for the photocatalytic activity of CeO_2_ nanoparticles to be improved by nano-assembly on graphene oxide-ZnO nanosheets. Recently, with this in mind, our associates have conducted a great deal of correlated research; for example, highly efficient photo-degradation was reported by R. Jeyagopal et al. [20] for the CoSnS@CNT hybrid catalyst using the hydrothermal method. Likewise, M. Karpuraranjith et al. used the hydrothermal method for enhancement of photocatalytic degradation with hexagonal SnSe nanoplate @ SnO_2_-CNTs [21] and better photocatalytic activity using the hydrothermal method was reported by B. Wang et al. [22] with an rGO wrapped trimetallic sulfide nanowire hybrid catalyst.

To focus on such problems, we used a cost-effective and facile hydrothermal strategy to synthesize GO-ZnO@CeO_2_ nanohybrids, and then investigated using several instrumental techniques such as HR-TEM, FT-IR, XRD, UV–visible DRS with band gap, N_2_-adsorption–desorption isotherm and mass spectroscopy. Furthermore, we conducted a systematic study of the degradation of methylene blue pollutant, which is effectively attributed to the typical nanosheets, interfacial contact, high surface area and band-gap matching.

## 2. Results and Discussion

### 2.1. Morphological Characterization

The surface morphological features that have been studied by HR-TEM performance are presented in Figure 1a–c. Fortunately, the recombination of semiconducting metal oxides (ZnO and CeO_2_) and graphene oxide by annealed and hydrothermal methods resulted in a unique well-defined ultrathin 2D nanohybrid layered structure with ZnO and CeO_2_ nanoparticles well dispersed on restacked structures of GO nanosheets. In addition, Figure 1d shows high-resolution TEM images of GO-ZnO@CeO_2_ nanostructures being established by selected area electron diffraction (SAED) patterns; the different crystallographic locations were seen in ZnO nanoparticles for (100), (002), (110) and (103), and CeO_2_ nanoparticles for (200) and (111), respectively [23]. Moreover, the ZnO nanoparticles develop an interlayer spacing of ~0.25 nm and the theoretical d-spacing indicates that the hexagonal crystal plane (002) and CeO_2_ make an efficiently porous layered assembly which has an interlayer distance ~0.31 nm and cubic crystal plane (111) [24]. The ultrathin layered structure of GO-ZnO@CeO_2_ may be favorable for the lower band gap and good surface activity concluded the photocatalytic reaction, further supported by the XRD results. In addition, the interfacial coupling between the ZnO@CeO_2_ nanostructure and graphene oxide sheet materials obviously plays a significant role in the photocatalytic efficiency.

### 2.2. Structural Characterizations

The functional group information of the CeO_2_, GO-ZnO, ZnO@CeO_2_ and GO-ZnO@CeO_2_ samples are shown in Figure 2. The FTIR spectrum of CeO_2_ displays several peaks at 3410, 2924, 2856, 1742, 1625, 1537, 1344, 1049, 958, 875 and 537 cm^−1^, which agrees with the characteristic chemical components in the stretching vibrations of O-H bonds and metal–oxygen bonds [25].

GO-ZnO nanoparticles exhibit various peaks that expose the existence of oxygen in the samples and additionally display several peaks at 1797, 1443 and 1048 cm^−1^ which correspond to the functional groups of carbonyls (C=O), tertiary (C-OH) and alkoxy (or) epoxy (C-O), respectively. The main characteristic peaks at 3431 cm^−1^ may be accredited to the stretching vibration of (O-H) hydroxyl groups. Furthermore, the obtained peak at 1630 cm^−1^ may denote the remaining sp^2^ bonds of graphene oxide [26]. The FT-IR spectrum of ZnO nanoparticles has a peak at 428 cm^−1^ that signifies the stretching vibration of metal–oxygen bonds.

The characteristic peaks of ZnO@CeO_2_ nanoparticles, compared to CeO_2,_ show few new peaks attributed to the assembly of both (ZnO and CeO_2_ peaks), shifted to 22 cm^−1^. It is interesting to note that GO-ZnO@CeO_2_ nanohybrids, compared to GO-ZnO, display some new peaks attributed to the mixture of CeO_2_ and the metal oxide; the peak seems to have shifted to ~47 cm^−1^. A strong peak at 422 cm^−1^ is attributed to the vibration of the Zn-O bond in ZnO and the peak at 475 cm^−1^ is attributed to the vibration of the Ce-O bond in CeO_2_, due to the interaction of Zn^2+^ and Ce^4+^ ions with graphene oxide. Therefore, the results obviously show that the co-existence of GO, an inorganic counterpart of metal oxides, can be an effective cover on a nanohybrid structured graphene oxide surface.

The structural information of CeO_2_, GO-ZnO, ZnO@CeO_2_ and GO-ZnO@CeO_2_ nanohybrids were investigated by X-ray diffraction analysis and their results are revealed in Figure 3. X-ray diffraction patterns of cubic structure CeO_2_ (JCPDS cars no. 34-0394) exhibit diffraction peak 2θ values at 28.6°, 33.5°, 46.7°, 56.8° and 68.7°, which is well indexed to the lattice planes corresponding to (111), (200), (220), (311) and (222), respectively [27]. The powder XRD pattern of GO-ZnO nanohybrids was obtained in the diffraction peaks at 2θ = 31.3°, 34.1°, 36.3°, 47.8°, 56.7°, 62.8° and 69.4° corresponding to the (100), (002), (101), (102), (110), (103) and (112) characteristic crystal diffraction planes of ZnO (hexagonal structure) as (JCPDS-36-1451) confirmed [28,29]. In addition, a low intensity diffraction peak was observed, the 2θ value at 26.4° well matching the lattice plane (002) of graphene oxide [30], confirming the existence of GO-ZnO nanostructures.

It is interesting to note that the X-ray diffraction patterns of ZnO@CeO_2_ and GO-ZnO@CeO_2_ nanohybrids, revealed in several diffraction planes of the cubic structure of CeO_2_ and hexagonal wurtzite structure of ZnO nanoparticles, were confirmed and no characteristic peaks of graphene oxide were detected, indicating the influence of metal oxide nanoparticles. Moreover, XRD analysis reveals a good crystalline phase for ZnO and CeO_2_ nanomaterials using the hydrothermal method, which is reliable through TEM analysis results.

### 2.3. Optical and Surface Area Characterization

UV–visible absorbance spectra and suitable band-gap matching studies were investigated for insight into the photocatalytic reaction. The absorbance spectrum of CeO_2_, GO-ZnO, ZnO@CeO_2_ and GO-ZnO@CeO_2_ nanohybrids are shown in Figure 4A. A broad absorption peak is noticed at 350 nm in the absorption spectra of CeO_2._ The absorption edges of GO-ZnO at 360 nm and GO-ZnO@CeO_2_ appearing at 356 nm were blue shifted towards a lower wavelength, which suggests a reduction in the band gap from the introduction of GO [31,32,33]. Furthermore, a characteristic absorption through a powerful transition in the UV region was detected in the above samples, exchangeable with the intrinsic band absorption of ZnO@CeO_2_ for electron shift from the valence band into the conduction band [34].

In Figure 4B, the optical energy gap (Eg) of the nanohybrids was determined via Tauc’s plots, α(hν) = A(hν − Eg)^n/2^. The estimated energy gaps of CeO_2_, GO-ZnO, ZnO@CeO_2_ and GO-ZnO@CeO_2_ nanohybrids were calculated to be 2.63, 2.92, 2.84 and 2.70 eV, respectively. Furthermore, lowering the band-gap energy of GO-ZnO@CeO_2_ is reliable through the larger level reactive in the photocatalytic reaction, and it might remain attributed to the building of interfacial contact between ZnO@CeO_2_ nanostructures and GO nanosheets.

The N_2_ adsorption/desorption isotherms for the GO-ZnO and GO-ZnO@CeO_2_ nanohybrids exhibit typical IV isotherm (at high P/P_0_) according to the IUPAC organization. As shown in Figure 5a,b, the BET surface area and pore volume for the GO-ZnO and GO-ZnO@CeO_2_ nanohybrids were calculated to be (23.8 m^2^/g and 0.166 cc/g) and (47.4 m^2^/g and 0.237 cc/g), respectively. These results are more helpful for enhancement of the photocatalytic activity.

### 2.4. Photocatalytic Dye Degradation

The photocatalytic performance of as-prepared CeO_2_, GO-ZnO, ZnO@CeO_2_ and GO-ZnO@CeO_2_ nanohybrid photocatalysts were examined using methylene blue dye by UV–visible spectroscopy. Investigating the presence of the photocatalyst, it can be seen that the dye absorption curve against time decreases in peak intensity under visible light conditions. Figure 6a, without photocatalyst, exhibits the MB dye absorption peak at 664 nm and no significant change is seen up to the 90 min interval. Correspondingly, introducing the nanocatalyst shows a large level decrease in the peak intensity. Photocatalytic performance was demonstrated by the self-degradation achieved (7.8%) by the MB dye. The photodegradation of the MB dye in the presence of a photocatalyst, for instance CeO_2_ (36.82%), GO-ZnO (57.43%) and ZnO@CeO_2_ (78.52%), might be degraded for 90 min. The obvious photodegradation effectiveness of GO-ZnO nanostructures was enhanced; the degradation efficiency achieved (96.66%) by the doping of CeO_2_ nanoparticles was larger than for the GO-ZnO and ZnO@CeO_2_ nanostructures. The photocatalytic activities of nanohybrids, after 90 min duration, were obtained in the order of GO-ZnO@CeO_2_ > ZnO@CeO_2_ > GO-ZnO > CeO_2_; there may be a combined effect caused by the graphene oxide making nano-structures on ZnO@CeO_2_ and possibly reducing the electron–hole recombination, enhancing the degradation efficiency.

Figure 6b, the photodegradation reactions of methylene blue, can be evaluated by using Langmuir Hinshelwood pseudo-first-order kinetics. Interestingly, the rate constant (k) for GO-ZnO@CeO_2_ is the fastest kinetics value at 2.416 min^−1^ and is 1.5 times higher than those of ZnO@CeO_2_ (1.603 min^−1^), GO-ZnO (0.924 min^−1^) and CeO_2_ (0.546 min^−1^), respectively.

Figure 6c exhibits the total percentage of photodegradation efficiency, completing the response on the 90 min interval, for the photocatalytic performance of MB dye (36.82, 57.43 78.52 and 96.66%) obtained with the corresponding nanohybrid photocatalyst of CeO_2_, GO-ZnO, ZnO@CeO_2_ and GO-ZnO@CeO_2_, respectively. Thus, the consequences suggest that the combination effect of GO-ZnO@CeO_2_ can effectively induce the photocatalytic activity of GO and ZnO@CeO_2_ through suitable band-gap matching and interfacial contact on the nanosheets. The achieved degradation efficiency outcomes were compared to other hybrid catalysts and shown in (Table 1) [35,36,37,38]; the GO-ZnO@CeO_2_ nanohybrid catalyst displayed highly effective photocatalytic results. Furthermore, the photocatalytic recyclability of the GO-ZnO@CeO_2_ nanohybrid catalyst was also examined and the results revealed in Figure 7a,b. The degradation activity of MB dye molecules demonstrates the four repeated cycles achieving about ~96.66 to 90.62% within 90 min, which suggests that GO-ZnO@CeO_2_ is confirmed to have good stability as well as recyclability for the degraded MB dye.

Pleasingly, the photodegradation mechanism is demonstrated in Figure 7c. An MB pollutant molecule in the aqueous solution and the GO-ZnO@CeO_2_ nanohybrid photocatalyst were treated using UV or visible light, and produced the electron–hole pairs. The produced holes and electrons are strong oxidizing and reducing agents, respectively. The produced electrons in the conduction band respond with O_2_ molecules to form a superoxide radical ion (*O_2_^−^). The produced holes in the valence band respond with H_2_O molecules, resulting in the creation of (*OH) hydroxyl radical. The graphene oxide acts as an electron–hole pair separation, to avoid their recombination, and a good electron acceptor. Lastly, the *OH radicals, which can mineralize the organic molecules and oxidize, respond with MB molecules, which results in the making of various species, such as CO_2_, water and other intermediate products [39,40,41,42], as in the following equations:

GO-ZnO@CeO_2_ + hυ (visible) → GO-ZnO@CeO_2_ (e_CB^−^_ + h_VB^+^_)(1)

2e_CB^−^_ +O_2_ → *O_2^−^_(2)

h_VB^+^_ + H_2_O → H^+^ + *OH(3)

MB + GO-ZnO@CeO_2_ → MB* + e_CB^−^_ (GO-ZnO@CeO_2_)(4)

MB* + *OH → intermediate products (CO_2_ + H_2_O + …)(5)

MB* + *O_2^−^_ → intermediate products (CO_2_ + H_2_O + …)(6)

To understand the intermediate product degradation of MB dye, different stages were further validated by mass spectroscopy. The test was controlled by using ESI-MS in the (+ve) ion scanning mode to detect the (*m*/*z*) mass-to-charge ratio of the samples. The mechanistic pathway of the photodegradation of MB dye to form intermediates using the GO-ZnO@CeO_2_ nanohybrid photocatalyst is shown in Figure 8a,b. The molecular weight of methylene blue (319.85) was detected, but the representative high-intensity peak was observed at 284 (*m*/*z*), due to the removal of (Cl^−^) ions [43,44]. In consequence of the degradation of MB dye, the photogenerated active species such as *OH could attack the essential carbon atom of MB to adeptly decolorize the dye molecule.

Noticeably, after 90 min of photodegradation, the constituents have gone; the preliminary step of the MB degradation would attack the *OH radicals that adapt the C–S ^+^=C and C=N–C bonds to the C–SH (=O)_2_–C and C–NH–C. The dye degradation intermediate products obtained at (*m*/*z*) 202, 137, 110, 104 and 74 were detected. Since the visual detection and investigative information, it is obvious that the MB dye has been completely degraded into different compounds via the cleavage of the (C-N) azo group bond. Hereafter, in the experimental procedure, the MB dye was oxidized and mineralized into small molecules, such as CO_2_, H_2_O and other intermediate constituents (Figure 1). The methylene blue degradation intermediate product confirmed the results and agreed with the previously described literature on dye degradation performance [45,46].

## 3. Experimental Details

### 3.1. Materials

Cerium nitrate (Ce (NO_3_)_3_·6H_2_O), zinc nitrate (Zn (NO_3_)_2_·6H_2_O), graphite powder, hydrochloric acid (HCl), hydrogen peroxide (H_2_O_2_), nitric acid (HNO_3_), sulfuric acid (H_2_SO_4_), sodium thiosulfate (Na_2_S_2_O_3_·5H_2_O), sodium nitrate (NaNO_3_) and potassium permanganate (KMnO_4_) were purchased from a Nice chemical factory, Mumbai, India. Methylene blue (C_16_H_18_ClN_3_S_23_H_2_O; MW 319.85) was obtained from the Merck chemical industry, Mumbai, India. All the chemicals and reagents have an AR grade and are used without any further process. All the experimental work also used Millipore water.

### 3.2. Preparation of Graphene Oxide (GO) Nanosheets

The graphene oxide nanosheet (GO) was synthesized from graphite powder by adapted Hummer’s technique; details were discussed in a previous report [26].

### 3.3. Synthesis of Ultrathin Layered GO-ZnO@CeO_2_ Nanohybrids

In a typical experiment, 250 mg of graphene oxide powder was homogenously dispersed in a 25 mL ethanol and water mixture. In addition, the solution was kept under ultrasonicate for 60 min to get a consistent solution at room temperature (A). Meanwhile, equal volumes of 0.5 M zinc nitrate and 0.5 M cerium nitrate solution were continuously stirred for 30 min to obtain a homogenous solution (B). Subsequently, A and B solutions were mixed in a single bath and stirred for 10 min to achieve a dispersal mixture, and newly arranged 25 mL of 1 M sodium hydroxide solution was added drop by drop into the dispersion mixture. After that, the reaction bath product was transferred to a 100 mL Teflon-lined stainless-steel autoclave and put into a hot air oven kept at 180 °C for 1 day. The resultant product was cooled down at room temperature and then washed through a water/ethanolic mixture several times. It was filtered in a suction pump and dried at 80 °C for 12 h. Finally, the obtained powder was annealed using a muffle furnace at 300 °C heating for 4 h and designated as GO-ZnO@CeO_2_ nanohybrids. In a similar way to prepared GO-ZnO without CeO_2_, ZnO@CeO_2_ was prepared without graphene oxide.

### 3.4. Characterizations

The functional groups of graphene oxide, zinc oxide and cerium oxide were characterized and identified by FT-IR spectroscopic analysis using Detector-TGS with KBr pellets (FT/IR-4600 type). Powder-XRD of the GO-ZnO@CeO_2_ nanostructures was approved with X-ray diffractometer (model XPERT-PRO). Morphological construction and shape distribution of the GO-ZnO@CeO_2_ nanohybrids were examined by JEOL-2100+ High Resolution Transmission Electron Microscopy (Accelerating Voltage: 200 kV). Optical behavior and suitable band-gap energy of GO-ZnO@CeO_2_ were measured by JASCO UV–visible NIR (V-670). The intermediate constituents of the degraded dye molecules were examined using JEOL GC MATE-II mass spectrometer. Specific surface area and pore volume of the nanohybrids were examined by N_2_- adsorption–desorption isotherm using (Quantachrome Instrument v11.05), Quantachrome Novawin, Boynton Beach, FL, USA.

### 3.5. Photo-Catalytic Experiments

In a distinctive test, the photocatalytic efficiency of synthesized nanohybrid catalyst was examined by methylene blue dye as a model contaminant. About 50 mg nanohybrid catalyst was well spread in 100 mL of MB dye (10 ppm) aqueous solution [21]. Before irradiation, the suspension mixture was magnetically stirred for 30 min to make the adsorption–desorption isotherm equilibrium. Subsequently, the reaction product was subjected to visible light irradiation (UV filter cut off at 400 nm with 350 W Xenon lamp from Juyuanguangdian Technology Co., Ltd., Chengdu, China). An amount of 3 mL of the treated solution was taken at different time intervals to analyze the concentration difference of organic dye molecules by UV–visible spectrophotometer in the range from 400 to 800 nm (maximum absorption peak at 664 nm using (INESA L5) from Zhejiang Scientific Instruments Co., Ltd., Zhejiang, China).

## 4. Conclusions

In summary, for the first time, a new GO-ZnO@CeO_2_ nanohybrid was synthesized using a hydrothermal method. A variety of techniques were used (FT-IR, XRD, HR-TEM, UV–visible DRS with band gap and BET analysis) to characterize the obtained materials. The GO-ZnO@CeO_2_ nanohybrids exhibit excellent photocatalytic performance for the degradation of organic pollutants (96.66%), under visible-light irradiated with favorable cycling stability. Furthermore, it is expected that the present work will offer new prospects for the design of low-cost nanohybrids with efficient photocatalytic activity and upcoming applications in environmental organization.

## Data Availability

Not applicable.

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
