# Peer review of "Hierarchical Ultrathin Layered GO-ZnO@CeO2 Nanohybrids for Highly Efficient Methylene Blue Dye Degradation"

_molecules, 2022, doi:10.3390/molecules27248788_

Round 1

Reviewer 1 Report

Authors presented the synthesis of nano-assembly on graphene oxide, zinc oxide and cerium oxide (GO-ZnO@CeO2) nanohybrid, which showed very interesting photocatalytic performance. All data and its interpretation look scientifically and technically appropriate.

I am wondering dispersibility of those particles in reaction medium where MB dyes and nanohybrids were reacted. As you know, the reactivity of nano-catalyst strongly depends on its surface area. This means highly aggregated nanohybrid in water has very low surface area which probably has very low catalytic performance.

I do not accept improved catalytic performance of your nanohybrid without providing their dispersibility and surface area.  

Reviewer 2 Report

In this manuscript, nano-assembly on graphene oxide, zinc oxide and cerium oxide nanohybrids were prepared, characterized, and tested. The nano-assembled structure of GO-ZnO@CeO2 exhibits a better performance (96.66 %) of MB dye degradation activity than GO-ZnO nanostructures and CeO2nanoparticles. This work provides some interesting results. However, in this present manuscript, the relationships between performance and structure of the samples were not so clear. Since the paper if conducted with great thoroughness, this manuscript seems publishable after a revision:

1.     Title ‘…highly efficient organic pollutant degradation’ seems inappropriate, for there was only MB dye degradation in the present work.

2.     Specific surface area of the samples and the adsorption results can be present if possible.  

3.     In Page 3 to 4: ‘In addition, the low intensity diffraction peak was observed 2θ value at 26.4° well matching the lattice plane (002) of graphene oxide [28], this is confirming the existence of GO-ZnO nanostructures.’ However, why the peak due to plane (002) was not detected in Fig.2(d)?

4.     In Page 4: ‘Moreover, XRD analysis exposes that the good crystalline phase ZnO and CeO2 nanomaterials turn to an amorphous phase through the hydrothermal method, which is dependable through TEM analysis results.’ Why the ZnO and CeO2 turn to amorphous phase? Or the crystallinity of the oxides reduced?

5.     Fig. 4 was provided incomplete.

6.     In Fig. 3(c), label ‘0.25 nm’ seems longer than that of ‘0.31 nm’, and the width '0.25nm' was not as 1/200 of the length of 50 nm. Please check.

7.     TEM images of samples (a) CeO2, (b) GO-ZnO, (c) ZnO@CeO2 should be provided if possible. Hierarchical ultrathin layered structure of the sample seems not clear enough.

Reviewer 3 Report

Yuanfu Chen et al. investigated the hierarchical ultrathin layered GO-ZnO@CeO2 nanohybrids for highly efficient organic pollutant degradation. The authors used the hydrothermal method to prepare the GO-ZnO@CeO2. The experiments and results are relatively completed. However, there are still some major issues to be solved. Therefore, I suggest a major revision.

1、  The subheading used in "Results and discussion" is too simple and does not serve the purpose of a title. Also, I think it would be more logical to put "Morphological characterization" before "Structural characterization".

2、  In Figure 1, the results of (a)CeO2 and (c)ZnO@CeO2, (b)GO-ZnO and (d)GO-ZnO@CeO2 should be put together for easier comparison.

3、  In Figure 2, authors should put all the XRD curves in one graph for easier comparison.

4、  In Figure 3, the authors characterized the TEM image of the sample and concluded that "it is efficiently porous layered assembly", but they did not explain the effect of its structure on the catalytic performance. Moreover, porosity of the samples is better to be characterized by BET.

5、  Figure 4B is incomplete, only half of the figure is shown.

6、  In Figure 6c, there is lack of experimental proof of band gap matching of ZnO and CeO2.

7、  The SAED pattern in Figure 3d needs to be discussed in detail.

8、  In Figure 5, the unit for y-axis was put in wrong position.

9、  The writing needs to be polished.

Reviewer 4 Report

This manuscript may be published after the following revisions:

1. In Figure 5, please add the adsorption results obtained at two different adsorption times, to show if the adsorption equilibrium has been reached? In addition, the k  and R2 values need to be provided in Figure 5(b).

2. The authors had better add some opto-electronic measurement (such as electrochemical impedance spectroscopy, transient photocurrent response, photoluminescence and electron paramagnetic resonance) to support the mechanism proposed in Figure 6(c). for this, the authors may refer to some works: https://doi.org/10.1016/j.seppur.2021.120161https://doi.org/10.1016/j.powtec.2020.01.009.

Round 2

Reviewer 1 Report

All my comments were satisfied, so I’d like to recommend publication of this study

Reviewer 2 Report

1. 'absorbed’ in the Label of Fig.5 was incorrect.

2.     Please check the accuracy of values of BET specific surface areas or the computation error in page 8.

Reviewer 3 Report

I am satisfied with the revision, and thus the manuscript can be published in current form.

Reviewer 4 Report

The revision is acceptable.
